# Novel Green Biosynthesis of 5-Fluorouracil Chromium Nanoparticles Using *Harpullia pendula* Extract for Treatment of Colorectal Cancer

**DOI:** 10.3390/pharmaceutics13020226

**Published:** 2021-02-06

**Authors:** Mohammed S. Saddik, Mahmoud M. A. Elsayed, Mohamed Salaheldin A. Abdelkader, Mohamed A. El-Mokhtar, Jelan A. Abdel-Aleem, Ahmed M. Abu-Dief, Mostafa F. Al-Hakkani, Hatem S. Farghaly, Heba A. Abou-Taleb

**Affiliations:** 1Department of Pharmaceutics and Clinical Pharmacy, Faculty of Pharmacy, Sohag University, P.O. Box 82524, Sohag 82524, Egypt; mohammed.sherif@pharm.sohag.edu.eg; 2Department of Pharmacognosy, Faculty of Pharmacy, Sohag University, Sohag 82524, Egypt; m.salaheldin@pharm.sohag.edu.eg; 3Department of Medical Microbiology and Immunology, Faculty of Medicine, Assiut University, Assiut 71516, Egypt; elmokhtarma@aun.edu.eg; 4Department of Industrial Pharmacy, Faculty of Pharmacy, Assiut University, Assiut 71516, Egypt; Jelan.abdelrazik@pharm.aun.edu.eg; 5Chemistry Department, College of Science, Taibah University, Madinah 42353, Saudi Arabia; amamohammed@taibahu.edu.sa; 6Chemistry Department, Faculty of Science, Sohag University, Sohag 82524, Egypt; 7Department of Chemistry, Faculty of Science, New Valley University, Al-Kharja 72511, Egypt; mostafa.f@scinv.au.edu.eg; 8Department of Biochemistry, Faculty of Pharmacy, Nahda University (NUB), Beni-Sueif 62511, Egypt; hatemsayed@rocketmail.com; 9Department of Pharmaceutics and Industrial Pharmacy, Faculty of Pharmacy, Nahda University (NUB), Beni-Suef 62511, Egypt; heba.elsayed@nub.edu.eg

**Keywords:** chromium nanoparticles, 5-Flourouracil, colorectal cancer, green biosynthesis

## Abstract

Colorectal cancer (CRC) is the third highest major cause of morbidity and mortality worldwide. Hence, many strategies and approaches have been widely developed for cancer treatment. This work prepared and evaluated the antitumor activity of 5-Fluorouracil (5-Fu) loaded chromium nanoparticles (5-FuCrNPs). The green biosynthesis approach using *Harpullia (H) pendula* aqueous extract was used for CrNPs preparation, which was further loaded with 5-Fu. The prepared NPs were characterized for morphology using scanning and transmission electron microscopes (SEM and TEM). The results revealed the formation of uniform, mono-dispersive, and highly stable CrNPs with a mean size of 23 nm. Encapsulation of 5-Fu over CrNPs, with a higher drug loading efficiency, was successful with a mean size of 29 nm being produced. In addition, Fourier transform infrared (FTIR) and X-ray diffraction pattern (XRD) were also used for the investigation. The drug 5-Fu was adsorbed on the surface of biosynthesized CrNPs in order to overcome its clinical resistance and increase its activity against CRC cells. Box–Behnken Design (BBD) and response surface methodology (RSM) were used to characterize and optimize the formulation factors (5-Fu concentration, CrNP weight, and temperature). Furthermore, the antitumor activity of the prepared 5-FuCrNPs was tested against CRC cells (CACO-2). This in vitro antitumor study demonstrated that 5-Fu-loaded CrNPs markedly decreased the IC50 of 5-Fu and exerted more cytotoxicity at nearly all concentrations than 5-Fu alone. In conclusion, 5-FuCrNPs is a promising drug delivery system for the effective treatment of CRC.

## 1. Introduction

Colorectal cancer (CRC) is the third highest major cause of morbidity and mortality worldwide [1]. Many strategies have been developed for cancer treatment, including surgery, chemotherapy, and radiation [2,3,4]. Despite advances in treatment strategies, resistance to chemotherapy represents a challenge in the management of the incurable metastatic disease [5].

The drug 5-Fluorouracil (5-Fu) is a well-known chemotherapeutic agent, used as a first-line treatment for CRC [6,7]. The drug represses DNA synthesis by hindering thymidylate synthetase [8]. However, the acquired drug resistance, short half-life of 10–20 min, and harmful side effects on the bone marrow and gastrointestinal tract (GIT), remain among the limitations of its clinical use [9,10]. Quite recently, considerable attention has been paid to using 5-Fu-based mix regimens and 5-Fu pro-drugs to conquer the clinical opposition and increase its anti-tumor activity [6].

In the last two decades, nanotechnology has gained great attention from medical, pharmaceutical, and chemical scientists because of its tremendously valuable uses in different fields. Nanoparticles (NPs) are popular drug delivery systems due to their controllable size, surface properties, and drug release dynamics [11,12,13,14,15,16].

Chromium (Cr) is a basic supplement associated with the guideline of carbohydrate and lipid metabolism [17]. It has been reported that the administration of Cr is associated with weight loss and diminishing body fat [18]. In addition, Cr, a naturally substantial metal, is widely used in electroplating, pigmenting, wood safeguarding, cowhide tanning, steel fabricating, material coloring, and paper and mash enterprises [19]. Alternatively, the drug has also been included or adsorbed on the outside of transporters. Nanoparticles have propelled research in drug delivery by facilitating changes in the dispersion of medications in the body and the delivery rate, expanding drug bioavailability and the porosity of the membrane [20,21,22,23]. However, studies on CrNPs are still lacking [24].

Green preparations of nanoparticles using plant extracts as capping and reducing agents are gaining popularity in the field of nanotechnology [25,26]. The reasons behind the use of this methodology include the fact it is centered around the use of environmentally friendly, financially savvy, and biocompatible lessening specialists for the combination of nanoparticles [27,28]. Members of the genus *Harpullia* are known to be rich sources of saponins, triterpenes, tannins, flavonoids, and sterols [29,30,31,32,33,34].

In this study, we used the aqueous extract of *H. pendula* in the green biosynthesis of CrNPs, that were then loaded with 5-Fu to yield improvements in its anti-cancer activity. As far as we know, this is the first study that has reported on the formulation of 5-Fu-loaded CrNPs for possible use in the treatment of CRC. The prepared NPs were characterized for size, morphology, FTIR, and XRD. In addition, the in vitro anti-tumor activity of 5-FuCrNPs against a colorectal cell line (CACO-2) was investigated.

## 2. Material and Methods

### 2.1. Materials

Chromium chloride (CrCl_3_·6H_2_O) and 5-fluorouracil (5-Fu) were purchased from Sigma-Aldrich (St. Louis, MO, USA).

The leaves of *H. pendula* were collected from the field of elaborate plants Faculty of Agriculture, Assiut University, Assiut, Egypt. All other solvents and reagents used in the study were of analytical grade and used as received.

### 2.2. Methods

#### 2.2.1. Plant Material and Extract Preparation

The air-dried powdered leaves of *H. pendula* (100 g) were mixed with 250 mL distilled water, and then the mixture was heated and boiled for 20 min to obtain the aqueous extracts. After cooling, the mixture was filtered using Whatman No. 1 filter paper and stored in the refrigerator until further use.

#### 2.2.2. Biosynthesis of Chromium Nanoparticles (CrNPs)

A total amount of 0.1 M of CrCl_3_·6H_2_O was prepared by dissolving 26.6 g of CrCl_3_·6H_2_O in 1 L of distilled water in a beaker with the aid of sonication. This was then kept until use. A total of 10 mL aqueous solution of *H. pendula* extract was mixed with 10 mL solution of CrCl_3_·6H_2_O at 35 °C. The mixture was stirred vigorously suing a magnetic stirrer for two hours then left in the dark for 24 h until the synthesis of CrNPs was complete. Later, the prepared chromium NPs were separated by centrifugation at 10,000 rpm for 10 min (Refrigerated Centrifuge. Model H3-20KR, Hunan Kecheng, Changsha, China). The supernatant was removed, and the residue was washed three times with deionized water. Finally, the precipitated CrNPs were dried in an oven for 1 h at 100 °C.

#### 2.2.3. Preparation of 5-Fu Loaded Chromium Nanoparticles (5-FuCrNPs)

Fifteen formulae of 5-FuCrNPs were prepared using Box–Behnken Design (BBD) [35,36]. Three independent variables were chosen: 5-Fu concentration (0.25, 0.5 and 0.75%), the weight of CrNPs (250, 500 and 750 mg), and reflux temperature in the water bath shaker (20 ± 1 °C, 25 ± 1 °C, and 30 ± 1 °C). CrNPs were added to previously prepared 200 mL aqueous solutions of different concentrations of 5-Fuin volumetric flasks (250 mL). They were then placed in a thermostatically controlled water bath shaker that was operated at different temperature degrees and 500 rpm for 24 h. Finally, the prepared 5-FuCrNPs were separated from the solution by ultrafiltration using a 50 kDa pore size filter [37].

#### 2.2.4. Characterization of the Prepared NPs.

Both CrNPs and 5-FuCrNPs were characterized using the following measurements.

##### X-ray Powder Diffraction (XRD)

XRD measurement was performed using a Philips X-ray diffractometer (PW 1710, Anode material Cu, at a voltage of 40 kV, current of 30 mA, Wavelength 1.541838 Å (Cu), Optics: Automatic divergence slit, Beta filtering using graphite, monochromator) for CrNPs. The samples were determined in the angle [2θ] ranging from 4.0° to 79° with increasing step by 0.06°.

##### Fourier Transform Infrared Spectroscopy (FTIR)

Infrared spectra of CrNPs and 5-FuCrNPs were examined using (Nicolet iS10 FTIR Spectrometer, Thermo Fisher Scientific, Waltham, MA, USA). Each sample was prepared, by mixing with 200 mg of KBr, followed by compressing under high pressure.

##### Scanning Electron Microscopy (SEM)

Each sample of solid CrNPs and 5-FuCrNPs was prepared by adhesion on double-adhesive carbon tape, which was coated with a gold thin film with a diameter of 150 to 200 Å. They were then scanned by a scanning electron microscope (JEOL model, Tokyo, Japan: JSM 5400LV) to investigate the structure and morphology of the sample.

##### Transmission Electron Microscopy (TEM)

Each sample of solid CrNPs and 5-FuCrNPs were prepared by dispersion in ethanol as a solvent before precipitating on the grid, and coating with a thin film of carbon. Finally, the samples were left to dry before being scanned by the transmission electron microscope (JEOL model: JEM-100 CXII., Tokyo, Japan). The particle size distribution of the prepared NPs was evaluated using image J Launcher, Tokyo, Japan, broken-symmetry software, version (1.4.3.6.7).

##### Determination of Drug Loading Efficiency Percentage (LE%)

The concentration of 5-Fu loaded on chromium NPs was spectrophotometrically determined by direct determination of unbound 5-Fu in the filtrate at λ_max_ 265 nm.

The amount of 5-Fu adsorbed by chromium nanoparticles q_e_ (mg/g) was calculated from the following Equation (1) [38]:q_e_ = (C_0_ − C_e_)V/m(1)
where C_0_ and C_e_ are the initial and equilibrium concentrations of 5-Fu. V is the volume of the 5-Fu (mL) and m is the Cr nanoparticles mass in mg.

The loading efficiency percentage (LE%) of 5-Fu adsorption was determined relative to the original drug concentration added, according to the following Equation (2):Loading Efficiency Percent (LE%) = [(C_0_ − C_e_)/C_0_] × 100(2)

#### 2.2.5. In Vitro Drug Release Study

In vitro release of 5-Fu from the formulated CrNPs was performed at 37 °C, predicted using a dissolution medium pH shift method with a paddle type dissolution test apparatus, SR II, 6 flasks (Hanson Research Co., Chatsworth, CA, USA) adjusted at 50 rpm [28,39]. In brief, 500 mL of simulated gastric fluid (pH 1.2) was used as the release medium for 1 h, followed by the addition of 5 mL of 7 M KH_2_PO_4_ containing 16.75% (*w*/*v*) NaOH in order to shift the pH to 7.4. The experiment then proceeded for an additional 2 h. Throughout the full experimental timeframe, a 3 mL aliquot was aspirated and filtered at every 30 min intervals to measure the absorbance at the foreordained λ_max_ of each media against a relating blank.

##### Experimental Design (BBD)

A Box–Behnken experimental design (BBD) was utilized to investigate and optimize the formulation parameters of 5-FuCrNPs preparation for maximum LE% and fast drug delivery after 1 and 3 h [40,41]. This design was used as it requires fewer treatment combinations in cases involving more than two dependent variables than are involved with other designs [42,43]. The BBD is also ratable and contains statistical “missing corners” which may be useful when the experimenter is trying to avoid combined factor extremes. This property prevents the potential loss of data in such cases [44]. A 3 factor 3 levels design was employed to design 5-FuCrNPs. The three independent formulation variables analyzed during the study were 5-Fu Concentration (X_1_), CrNP weight (X_2_), and temperature (X_3_). The selected factors with the actual and coded levels, as per the design, are represented in Table 1. According to this design, 15 formulae of 5-FuCrNPs were prepared. Three levels of 5-Fu concentration were used, 0.25, 0.5, and 0.75%, denoted by the values −1, 0, and +1, respectively, in the above design. The different CrNP weights used were 250, 500, and 750 mg, also denoted the values −1, 0, and +1, respectively. Lastly, the reflux temperature was chosen to be 20 ± 1 °C, 25 ± 1 °C, and 30 ± 1 °C denoted −1, 0, and +1 values, respectively. The dependent variables to be tested for the prepared 5-FuCrNPs were chosen to be the loading efficiency (Y_1_), release at the end of 1 h (Y_2_), and release at the end of 3 h (Y_3_).

#### 2.2.6. Effect on Cell Proliferation

The effect of 5-Fu, CrNPs, and 5-FuCrNPs on the viability of the CACO-2 cell line (Colorectal adenocarcinoma cell line, kindly supplied from VACSERA tissue culture lab, Obour City, Cairo, Egypt) was investigated using the MTT cell proliferation assay (Cell Titer 96^®^ Non-Radioactive Cell Proliferation Assay, Promega, Dane County, WI, USA) according to the manufacturer’s instructions. In brief, 5000 cells were plated in wells of 96-well plate in triplicates and incubated at 37 °C in a 5% CO_2_ incubator (Thermo Fisher Scientific, Waltham, MA, USA) for 24 h. On the next day, different concentrations of 5-Fu, CrNPs, and 5-FuCrNPs were added to the cells and treated cells were incubated for 24 h. Control cells were treated with PBS. After incubation, the MTT reagent was added to the cells and incubated for 4 h in the dark, followed by the addition of the stop solution to dissolve the formed crystals. The absorbance of the solution was measured at wavelengths of 570 and 630 nm (reference filter). Cell viability was expressed as the mean percentage of viability and was calculated by dividing the optical density of the treated sample/optical density of the control samples. The IC50 (the concentration of the sample that caused a 50% inhibition of cell viability) was determined from the nonlinear regression curve obtained by plotting the log concentration of the inhibitor vs. cell viability as a percentage. Each concentration of the inhibitor was tested in triplicate. Data are presented as the average IC50 for the tested inhibitory material. Growth curves and regression analysis were performed using GraphPad Prism 8.4 (GraphPad Software, San Diego, CA, USA). 

In addition, the ability of 5-Fu, and 5-FuCrNPs to induce apoptosis of the CACO-2 cells was analyzed by flowcytometry using annexin V and a propidium iodide staining kit (Thermo Fisher Scientific, Suwanee, GA, USA). Cells were acquired on FACSCalibur™ flow cytometer (BD Biosciences, San Jose, CA, USA) and data were analyzed by FlowJo software 8.7 (Treestar, Ashland, OR, USA).

## 3. Results and Discussion

In the current study, CrNPs were successfully produced using an eco-friendly green synthesis approach (Figure 1) [15]. The biosynthesis of CrNPs was carried out using *H. pendula* aqueous extract as the reducing and capping agent. Additionally, the *H. pendula* plant has been reported to be a rich source of phenolics and flavonoids, with total contents of 255.5 ± 7.18 mg gallic acid equivalents/g extract, and 111.6 ± 3.2 mg quercetin equivalents/g extract, respectively [33]. Different flavonoids and polyphenolic compounds have previously been isolated and characterized from the *H. pendula* plant, including rutin, vitexin, isovitexin, orientin, quercetin, and kaempferol quercetin-3-*O*-β-d-glucoside, kaempferol-3-*O*-β-d-glucoside, kaempferol-3-*O*-β-d-glucopyranosyl-(1→2)-α-l, rhamnopyranoside, kaempferol-3-*O*-β-d-glucopyranosyl-(1→4)-α-l-rhamnopyranoside, kaempferol-3-*O*-β-d-apiofuranosyl-(1→2)-β-d-glucopyranoside, and kaempferol 3-*O*-(6″galloyl) apiofuranosyl (1‴→2″)-β-galactopyranoside [31,33]. Additionally, members of this family have been widely studied for their antioxidant, anti-inflammatory, insecticidal, and anti-diabetic properties [45,46,47,48].

In the present work, the biosynthesized chromium nanoparticles were further loaded with 5-Fu and their anti-tumor activity was evaluated against the CACO-2 cell line. The biosynthesized CrNPs and 5-Fu-loaded chromium NPs were further characterized as described in the following subheadings.

### 3.1. XRD Diffraction of the Biosynthesized CrNPs

The spectrum of XRD comprises three clearly distinguishable peaks. They can all be indexed exactly, not only in the peak position, but also in their relative intensity to crystalline CrNPs. Peaks with 2θ values of 25.39°, 53.21°, and 73.30° correspond to the crystal planes of crystalline CrNPs (111), (200) and (311), respectively [24]. Scherer’s formula (Equation (3) can be used to predict the crystallite sizes [49,50,51,52].
*D* = *kλ*/*β cosθ*(3)
where *λ* is the X-ray wavelength, the constant *k* is 0.94, and *θ* and *β* are half of the Bragg angle and the half-width of the peak, respectively. By using Equation (3), the crystallite sizes were estimated to be approximately 25.5 nm (Figure 2a).

Analyzing the X-ray peak profile of the prepared CrNPs with the Williamson-Hall (W-H) method allows the crystallite size (i.e., coherently diffracting domains) and the lattice strain (i.e., the lattice parameters resulting from crystal imperfections) to be determined by observing the peak width as a function of the 2-Theta angle [53,54]. The corrected expansion of a peak *β_Correct_* is connected to the observed expansion *β_Correct_* and the instrumental expansion *β_Instrumental_* by the following relation (Equation (4)), according to the Lorentzian distribution feature.
(4)$$\beta_{Correct}=\beta_{Observed}-\beta_{Instrumental}$$

To measure the pure extension of the XRD lines of our investigated CrNP samples, the observed line integral breadth was used. After the Lorentzian fitting of the spectral XRD peaks in Figure 2b, the line integral breadth for a specific peak is calculated from the integral normalized by the maximum intensity of that peak. As a standard reference material (SRM), the XRD profile of a strongly crystalline LaB6 (660a) was used to measure the instrumental broadening correction in which the expansion effect due to particle size is small. The broadening of the diffraction peaks is attributable to crystallite size (*β_D_*) and micro-strain (*β_ε_*) contributions, according to the (W-H) approach.

Thus, the pure peak broadening (*β_Correct_*) for the sample in the understudy read (Equations (5) and (6)) [55]:(5)$$\beta_{Correct}=\beta_{D}-\beta_{\varepsilon }=\frac{k\lambda }{D\;cos\theta }+4\varepsilon \;tan\theta$$
(6)$$\beta_{Correct}\;cos\theta =\frac{k\lambda }{D_{L}\;}+4\varepsilon_{L}\;sin\theta$$
where *k* is the form factor (~0.94), *λ* is the XRPD wavelength of the incident (~0.15418 nm), *D_L_* is the mean size of the crystallite, and *ε_L_* is the average micro-strain. Equation (6) suggests the isotropic existence of the sample and the micro-strain is identical in all crystallographic (*hkl*) planes. This equation is the uniform model of deformation (UDM), in which the crystal is isotropic in nature [54,55]. For the prepared CrNPs, the average crystallite size (*D_L_*) and average micro-strain (*ε_L_*) were determined using Equation (4). Equation (4) is the linear regression between (*β_Correct_ cosθ*) and (4 *sinθ*), whereas as shown in Figure 2b, the *D_L_* (27.45 nm) and *ε_L_* (10.24 × 10^−3^) can be determined from the intercept and slope.

### 3.2. SEM and TEM Analyses of Biosynthesized CrNPs and 5-FuCrNPs

In the microstructural characterization studies, the size of the nanoparticles was determined, and the homogeneity and size distribution were examined. The particles were found to be nearly spherical, monodispersed, and smooth on the surface (cf. Figure 3a–c). In order to distinguish the morphology, scale, and shape of the nanoparticles, the biosynthesized CrNPs were also characterized by TEM. TEM images of the synthesized CrNPs were measured in the 23 nm range (Figure 4), where the 29 nm range of 5-FuCrNPs was measured (Figure 5). TEM images, with the distributions of particle size are shown by the histogram curves (Figure 4 and Figure 5). It can be inferred that in all cases, the spherical CrNPs demonstrated irregular morphologies and poly-dispersed characteristics with a nano-size up to 23 nm. The data of particle size obtained from TEM are correlated with the XRD analysis in Figure 2a.

### 3.3. Functional Groups Identification Utilizing FTIR Spectroscopy

In order to determine the purity and nature of the Cr nanoparticles and to confirm the encapsulation of 5-Fu over the CrNPs, FTIR spectroscopy was carried out (Figure 6). The peak between 3346 and 1702/cm is due to the -OH stretching and bending vibrations of adsorbed water molecules in the sample [56]. At 2383/cm, the characteristic absorption peak may have arisen from the primary amines present in the aqueous extract of the *H. pendula* [56]. The band at 1733/cm indicates evidence for the ester group [57]. The two absorption peaks observed at around 570/cm may be due to the partial amine or carboxyl group deutrization. The absorption peak is characteristic of OH plane bending at 1388/cm [58]. The peak at 2930 and 2860/cm may be due to –CH_2_ stretching vibrations [59]. C-H stretching in the long aliphatic chain of the fatty acid moiety is associated with peaks at 1442 and 762/cm [59]. The peaks appearing at 1727 and 1370/cm denote the C=O and C-O stretching vibrations, respectively, especially for 5-FuCrNPs. As per the aforementioned discussion, N-H, O-H, A-OH hydroxyl, aliphatic C-H stretching bonds are the peaks formed in the extract that are responsible for reducing Cr(III) to CrNPs in these extracts. These biomolecules can act as agents of reduction and stabilization [24].

### 3.4. Drug Loading Efficiency on CrNPs (LE%)

The effects of independent variables on the LE%, and drug release after 1 and 3 h are listed in Table 1. Regarding the 5-Fu concentration, the calculated theoretical drug content value for N1, N5, and N12 was 20%, while for N2, N3, N7, N8, N9, N13 and 14 it was 33.33% and for N4, N10, N11, and N15 it was 42.85%. The effect of changing the 5-Fu concentration on the LE% was investigated, at each 5-Fu concentration, CrNP weight and temperature, using the ANOVA test. The LE% depends on the 5-Fu concentration, CrNP weight and temperature, and ranges from 55.87 ± 2.03% to 92.11 ± 3.23%. The regression Equation (7) of the fitted LE% model is [36,42]: LE% = 79.45 + 7.21 X_1_ + 6.45 X_2_ − 4.99 X_3_ − 1.19 X_1_^2^ − 5.24 X_2_^2^ + 2.25 X_3_^2^ − 1.31 X_1_X_2_ + 0.57 X_1_X_3_ − 0.14 X_2_X_3_(7)

Since the *p*-value in the ANOVA table is less than 0.1, there is a statistically significant relationship between the variables at the 90% and higher confidence level. The R-squared statistic indicates that the model (as fitted) explains 36.23% of the variability in 5-Fu concentration. However, the output shows the results of fitting a multiple linear regression model to describe the relationship between CrNP weight and the LE%. Since the *p*-value is less than 0.1, there is a statistically significant relationship between the variables at the 90% confidence level. The R-squared statistic indicates that the model (as fitted) explains 37.87% of the variability in CrNP weight. The temperature had a non-significant effect on the LE%, as *p*-value is greater than 0.1, in spite of the decrease in LE% with the increase in temperature.

The 3D plot (Figure 7) showed that the LE% increased from 55.87 ± 2.03 to 74.32 ± 2.08 and from 80.25 ± 2.21 to 92.11 ± 3.23 at lower and higher 5-Fu concentrations with a constant CrNP weight and temperature. The improvement in LE% with increased speed levels was due to the fact that more of the drug is adsorbed and incorporated on the surface of CrNPs with a high 5-Fu concentration. A linear relationship between the two variables investigated the percentage drug incorporation. The biosynthesis of 5-FuCrNPs involves two major steps, the formation of stable CrNPs of the CrCl_3_ solution and the subsequent adsorption of 5-Fu on the surface of the CrNPs. These two steps have a significant effect on the size and LE% of nanoparticles prepared. Similar results were obtained for the preparation of biodegradable sorafenib-loaded carbon nanotubes for the treatment of hepatocellular carcinoma (HCC) [39].

Alternatively, the LE% increased from 75.64 ± 3.98 to 85.89 ± 2.01 and 74.32 ± 2.08 to 87.54 ± 2.02% at lower and higher CrNP weights with a constant 5-Fu concentration and temperature (Table 1). As the weight of the CrNPs increased, the surface area available for drug loading was increased. This increase in surface area allows more drug to be adsorbed on the surface of the nanoparticles.

The temperature degrees selected during nanoparticle preparation had a noticeable effect on the LE% of the drug; as the temperature increased, the LE% decreased. These results were expected, as the adsorption process of the drug on the surface of the CrNPs decreased with increasing the temperature of the reflux process of 5-Fu with CrNPs [60]. The main target of studying the temperature effect was to achieve the optimum reflux temperature that yields a higher LE%. The LE% decreased from 75.64 ± 3.98 to 67.31 ± 2.23% (N2, N3) and from 80.25 ± 2.21 to 67.76 ± 3.05% (N5, N6) at lower and higher levels of temperature with a constant 5-Fu concentration and CrNPs weight (Table 1). All these results confirm that to obtain higher values of LE%, a lower temperature should be used (20 ± 1 °C, −1).

### 3.5. In Vitro Release Study of 5-FuCrNPs

Regression Equations (8) and (9) show the output of the relationship between 5-Fu concentration, CrNP weight, and temperature with the percentage drug released after 1 and 3 h.
Rel 1 h = 16.10 + 3.365 X_1_ − 2.420 X_2_ + 0.527 X_3_ − 3.38 X_1_^2^ + 2.31 X_2_^2^ + 1.27 X_3_^2^ − 0.21 X_1_X_2_ + 0.19 X_1_X_3_ − 1.18 X_2_X_3_(8)
Rel 3 h = 84.73 + 7.37 X_1_ − 2.27 X_2_ − 1.15 X_3_ − 5.35 X_1_^2^ − 7.47 X_2_^2^ + 2.86 X_3_^2^ − 3.37 X_1_X_2_ + 2.06 X_1_X_3_ − 5.05 X_2_X_3_(9)

The 5-Fu concentration, as well as CrNP weight, has a significant effect on drug release (*p*-value < 0.1), after 1 and 3 h. The drug release percentage increases with an increase in their levels. In contrast to its effect on the LE%, the temperature had a non-significant effect on drug release.

Table 1 and Figure 8 and Figure 9 show the response surface plots of the in vitro release of 5-Fu from its CrNPs. The results show the in vitro release of 5-Fu from formulae (N1, N2, N3, and N4) using CrNPs (X_2_) at a constant level (−1) with variable 5-Fu concentrations (X_1_), 0.25% for N1; 0.5% for N2, N3, and 0.75% for N4; (X_2_). The maximum and minimum percent released was observed to be 14.88 ± 1.88 and 24.12 ± 1.77% at the end of 1 h (Y_2_). The maximum and minimum in vitro release after 3 h (Y_3_) of dissolution was 90.77 ± 1.72 and 62.15 ± 1.96%, respectively. These investigated formulae can be arranged, in descending order, concerning the in vitro release within 3 h as follows: N3 > N2 > N4 > N1, respectively.

The data for the in vitro release of 5-Fu from the CrNPs containing formulae N5, N6, N7, N8, N9, N10, and N11 using constant CrNP weight (X_2_) at the medium level (0), variable temperature levels (ranging from −1 to +1), and with variable 5-Fu concentrations (X_1_) are shown in Table 1. The maximum and minimum percentages released were 18.33 ± 1.43 and 11.89 ± 1.98% at the end of 1 h (Y_2_). The maximum and minimum in vitro release after 3 h of dissolution (Y_3_) were found to be 92.73 ± 1.33 and 76.09 ± 1.94%, respectively. According to these results, these formulae can be arranged, in descending order, concerning the in vitro release within 3 h dissolution as follows: N11 > N10 > N7 > N8 > N9 > N5 > N6, respectively. 

The in vitro release of 5-Fu from the CrNPs containing formulae N12, N13, N14, and N15 was examined using constant CrNPs weight (X_1_) at the medium level (0), variable temperature levels (ranging from −1 to +1), and with variable 5-Fu concentrations (X_1_).

The significant negative effect of CrNP weight on drug release after 1 and 3 h is obvious at different formulations, as revealed by Equations (8) and (9). At a constant 5-Fu concentration (X_1_) and temperature (X_3_) (0, 1), the percentage of 5-Fu released was 24.12 ± 1.77 and 18.61 ± 1.66% (N3, N14) at the end of 1 h (Y_2_) at lower and higher CrNP weights. The maximum and minimum in vitro release after 3 h of dissolution (Y_3_) was found to be 90.77 ± 1.72 and 69.93 ± 1.54%. The same findings were shown for N4 and N15 at lower and higher CrNP weights, as the percentage was released after 1 h decreased from 21.90 ± 1.65 to 15.92 ± 1.99%, and after 3 h from 79.42 ± 1.33 to 74.18 ± 1.63%.

All these results revealed the significant effect of increasing the 5-Fu concentration on the release of the drug after 1 and 3 h. These findings are correlated to the ease of drug release as it adsorbed on the surface of CrNPs and a greater amount of the drug is available for release with increasing its concentration. The amount released after 1 h was small compared to the amount released after 3 h. These results may be due to the difficulty of desorption of 5-Fu from CrNPs in the acidic environment and the nature of the prepared 5-FuCrNP system, which is still intact at a pH of 1.2. At an alkaline pH, the drug undergoes fast desorption behavior, and the amount released increased dramatically due to the basic nature of the 5-Fu and coated CrNPs that allows for a greater amount of the drug to be available for absorption sites. These results are valuable for decreasing the incidence of gastritis and the other 5-Fu side effects by decreasing drug release in the stomach. In addition to increasing drug bioavailability at absorption sites in the intestine, the green biosynthesis of CrNPs decreases Cr toxicity and enhances its clearance from the body by conjugation with bile salts.

The percentage of drug released was inversely proportional to CrNP weight. These results are in accordance with the fact that an increase in CrNP surface area with increasing NP weight allows more drug to be adsorbed on the surface on the monolayer pattern. This monolayer attachment of 5-Fu exerts a negative effect on the release of the drug from the CrNP surface. In contrast, as the CrNP weight was decreased, the amount of drug adsorbed in a multilayer pattern on the surface of CrNPs that makes it more easily released due to attachment forces was decreased. 

In contrast to LE%, temperature had a non-significant effect on 5-Fu release after 1 and 3 h. The results of LE% on in vitro release after 1 and 3 h revealed that the optimum selected formula of 5-FuCrNPs should be prepared at a high level of 5-Fu concentration (+1, 0.75%), medium level of CrNP weight (0, 500 mg) and low level of temperature (−1, 20 ± 1 °C). According to the rank order of all formulae (considering the higher LE% and Rel. 1 and 3 h), the optimum formula is N10, while the remaining formulae ranked as follows: N11, N3, N13, N8, N2, N4, N7, N15, N9, N14, N5, N12, N1, and finally, N6. The optimum formula N10 (+1, 0, and −1 for X_1_, X_2_, and X_3_) was selected for further cell viability study and antitumor activity study on the colorectal cell line (CACO-2).

### 3.6. In Vitro Cytotoxicity Assay

The cytotoxic effect of 5-FuCrNPs on CACO cells was investigated using the MTT assay and this was compared with that of 5-Fu alone. Both 5-Fu and 5-FuCrNPs exerted a dose-dependent cytotoxic effect (Figure 10a). Interestingly, 5-FuCrNPs markedly decreased the IC50 of 5-Fu from 106 to 19.7 µg/mL. Supporting our data, flow cytometry analysis showed that when cells were treated with 5-FuCrNPs (30 µg/mL), more late apoptotic cells were observed (42.7%) compared to cells treated with the same concentration of 5-Fu (20%) or CrNPs alone (11.7%) (Figure 10b–d).

The toxicity of chromium is attributed to its conversion to chromium (III) by OH· after passing through the ion channels of the cytoplasmic membrane generation [61]. Our results demonstrate that the toxicity of 5-FuCrNPs in CACO cells was dose-dependent and achieved better results than 5-Fu alone, as measured by the cellular reduction in MTT [62,63].

Similarly, chromium oxide nanoparticles exert significant cytotoxic activity on murine fibrosarcoma cells (L929) [63]. Chromium can induce cytotoxicity by different mechanisms. Xia, et al. [64] showed that cells treated with chromium suffer from increased DNA damage, such as mutations, including double- and single-strand breaks at minute damage sites of DNA [62]. In addition, exposure to chromium nanoparticles has been associated with enhanced intracellular reactive oxygen species and oxidative stress, pointing to the induction of apoptosis. Thia was also associated with decreased glutathione levels and lipid peroxidation [65,66]. Interestingly, treatment with chromium resulted in elevated expression of autophagy-related proteins such as Beclin1 and LC3II [67].

## 4. Conclusions

Chromium nanoparticles were prepared using a novel green biosynthesis methodology and successively loaded with 5-Fu. Photographs of SEM and TEM images revealed that the prepared 5-FuCrNPs were spherical in shape, with a smooth surface and have an optimum nanosized range for drug delivery to tumor tissues. All formulation parameters (5-Fu concentration, CrNP weight and temperature) had significant effects on the LE%. CrNPs loaded with 5-Fu could be a promising agent for the effective treatment of colorectal cancer as we demonstrated that they markedly decreased the IC50 of 5-Fu and exerted greater cytotoxicity than 5-Fu alone. Based on our interesting findings, we recommended the future use of the 5-FuCrNP combination to overcome clinical resistance and decrease the toxic effects on the bone marrow that are associated with a high dose of 5-Fu.

## Figures and Tables

**Figure 1 pharmaceutics-13-00226-f001:**
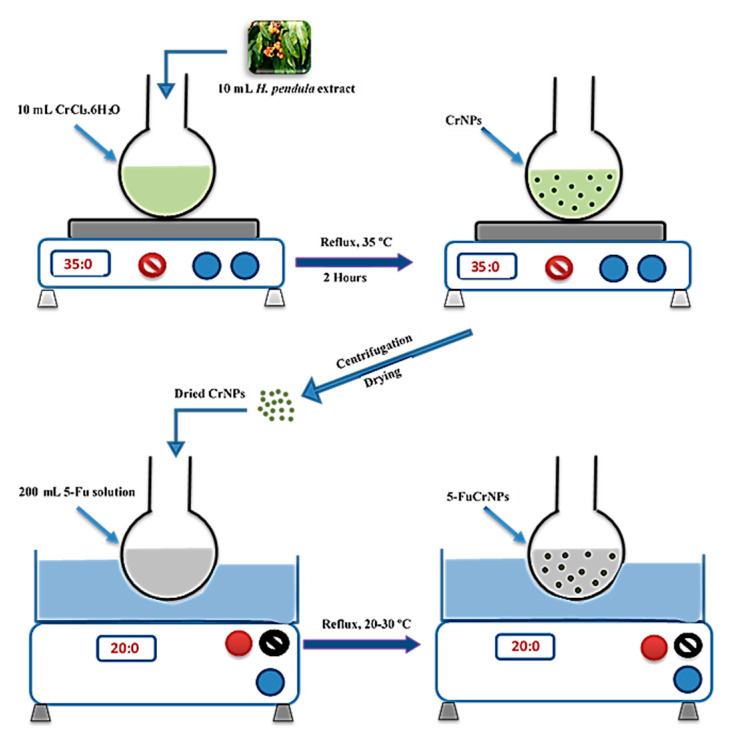
Schematic diagram showing the details of green biosynthesis preparation method, where, (CrNPs) is Chromium nanoparticles, (5-Fu) 5-Fluorouracil, and (5-FuCrNPs) 5-Fluorouracil chromium nanoparticles.

**Figure 2 pharmaceutics-13-00226-f002:**
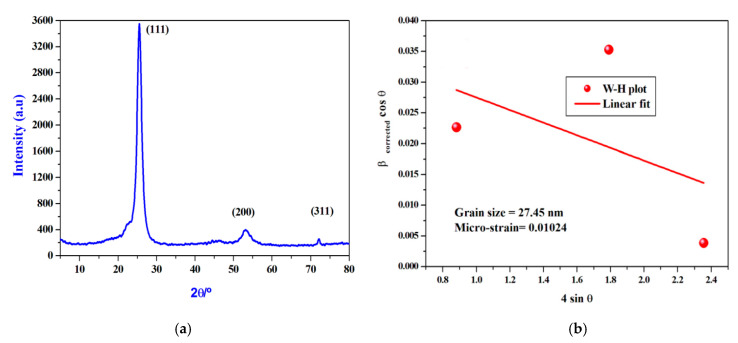
(**a**). X-ray powder diffraction (XRD) patterns of the prepared CrNPs. (**b**). The Williamson-Hall (W-H) plots for CrNPs. The calculations of (*β_correct_*) were carried out according to Psd-Voigt distribution functions.

**Figure 3 pharmaceutics-13-00226-f003:**
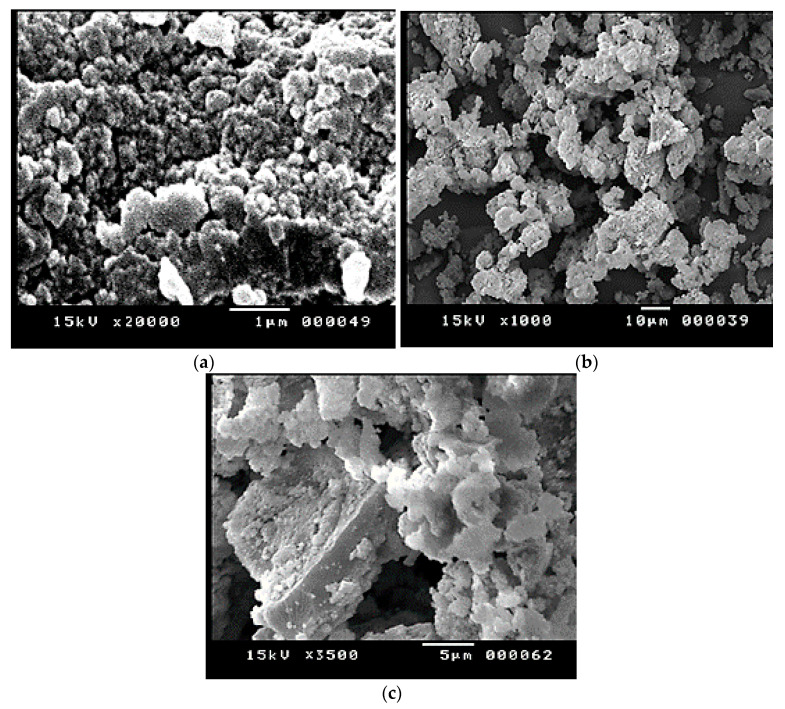
Scanning electron micrographs of the prepared (**a**) CrNPs, (**b**) 5-FuCrNPs N1 and (**c**) N10.

**Figure 4 pharmaceutics-13-00226-f004:**
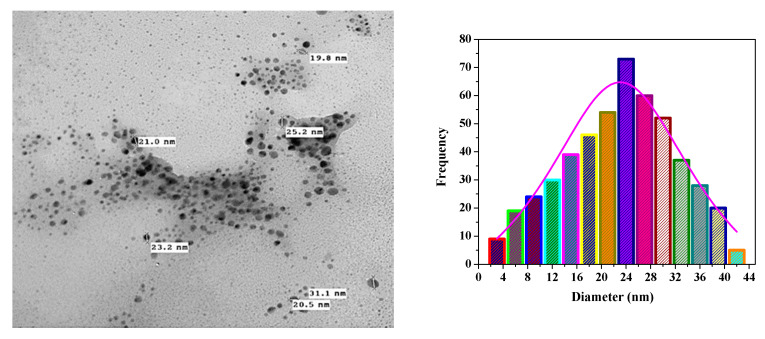
Transmission electron microscope analysis of the prepared CrNPs with a diameter distribution histogram.

**Figure 5 pharmaceutics-13-00226-f005:**
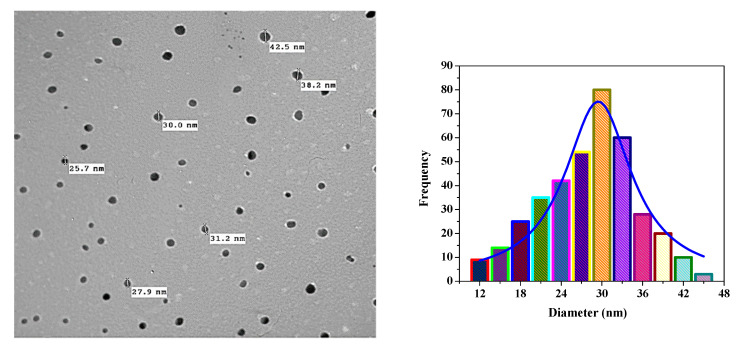
Transmission electron microscope analysis of the prepared 5-FuCrNPs with a diameter distribution histogram.

**Figure 6 pharmaceutics-13-00226-f006:**
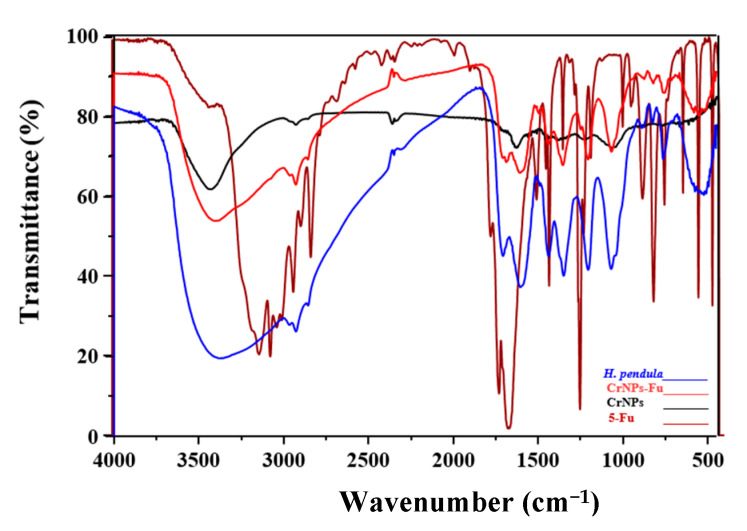
Fourier transform infrared spectra of the prepared CrNPs, 5-FuCrNPs, *H. pendula* extract, and 5-Fu.

**Figure 7 pharmaceutics-13-00226-f007:**
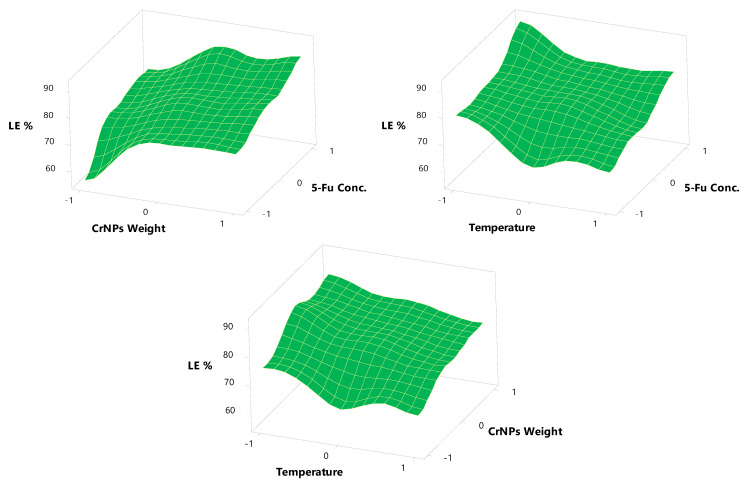
Surface plots effects of different formulation factors on LE%.

**Figure 8 pharmaceutics-13-00226-f008:**
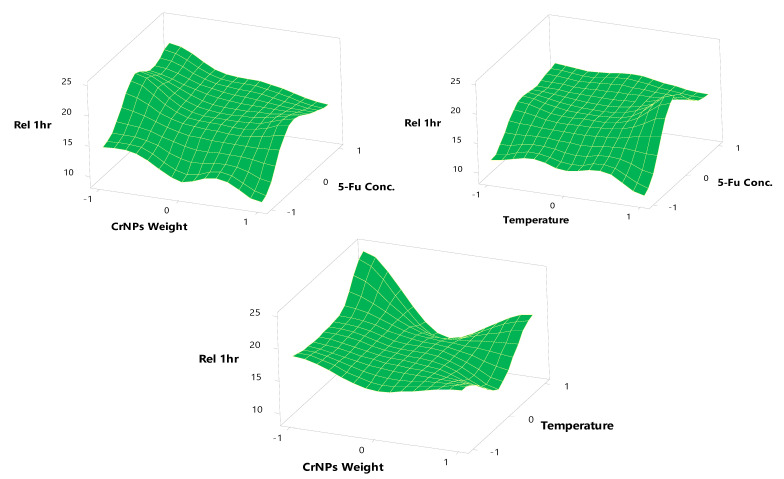
Surface plots effects of different formulation factors on Release after one h.

**Figure 9 pharmaceutics-13-00226-f009:**
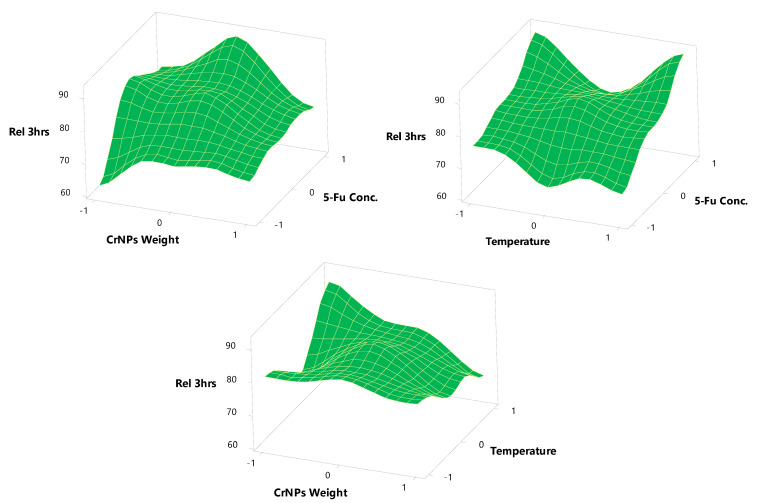
Surface plots effects of different formulation factors on release after 1 h.

**Figure 10 pharmaceutics-13-00226-f010:**
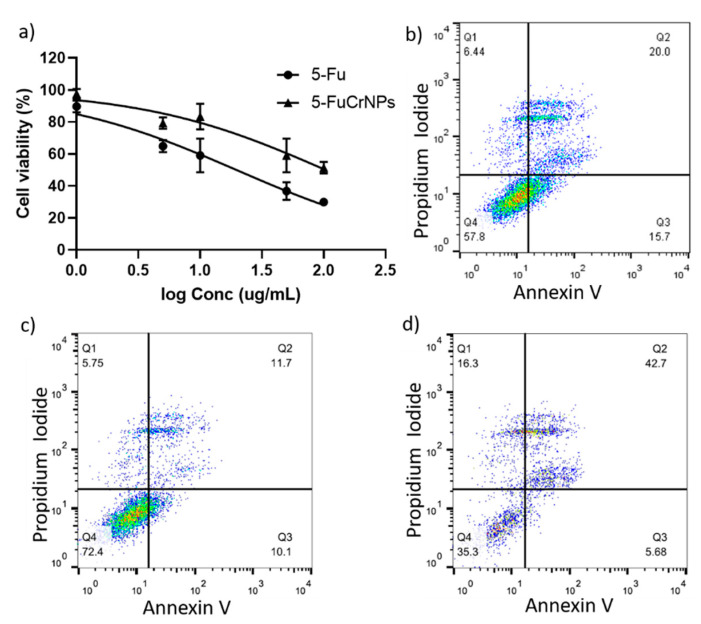
Viability of the CACO cells treated with different concentrations of 5-Fu and 5-FuCrNPs determined by the MTT assay (**a**). Flow cytometry analysis showing the frequency of apoptotic cells treated with 5-Fu (**b**), CrNPs (**c**), and 5-FuCrNPs (**d**).

**Table 1 pharmaceutics-13-00226-t001:** Observed values of responses for 5-fluorouracil chromium nanoparticles (5-FuCrNPs).

Formula No.	The Variable Level in a Coded Form			LE%	Cumulative Percentage Released	
	X_1_ 5-Fu Conc.	X_2_ CrNP Weight	X_3_ Temperature	Y_1_ LE %	Y_2_ Rel 1 h	Y_3_ Rel 3 h
N1	−1	−1	0	55.87 ± 2.03	14.88 ± 1.88	62.15 ± 1.96
N2	0	−1	−1	75.64 ± 3.98	18.26 ± 1.93	81.77 ± 1.11
N3	0	−1	1	67.31 ± 2.23	24.12 ± 1.77	90.77 ± 1.72
N4	1	−1	0	74.32 ± 2.08	21.90 ± 1.65	79.42 ± 1.33
N5	−1	0	−1	80.25 ± 2.21	11.89 ± 1.98	76.09 ± 1.94
N6	−1	0	1	67.76 ± 3.05	9.83 ± 1.34	68.13 ± 1.56
N7	0	0	0	74.13 ± 2.06	18.14 ± 1.65	87.11 ± 1.65
N8	0	0	0	83.23 ± 4.12	14.24 ± 1.74	84.51 ± 1.44
N9	0	0	0	80.99 ± 3.11	14.02 ± 1.83	82.93 ± 1.76
N10	1	0	−1	92.11 ± 3.23	18.33 ± 1.43	91.23 ± 1.34
N11	1	0	1	81.91 ± 3.87	16.36 ± 1.87	92.73 ± 1.33
N12	−1	1	0	74.33 ± 2.99	9.88 ± 1.96	71.45 ± 1.93
N13	0	1	−1	85.89 ± 2.01	16.02 ± 1.54	80.69 ± 1.25
N14	0	1	1	76.81 ± 3.43	18.61 ± 1.66	69.93 ± 1.54
N15	1	1	0	87.54 ± 2.02	15.92 ± 1.99	74.18 ± 1.63

## Data Availability

Not applicable.

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
