# Peer review of "Novel Green Biosynthesis of 5-Fluorouracil Chromium Nanoparticles Using Harpullia pendula Extract for Treatment of Colorectal Cancer"

_pharmaceutics, 2021, doi:10.3390/pharmaceutics13020226_

Round 1

Reviewer 1 Report

The authors investigate 5-Furouracil (5-Fu) loaded nanoparticles from biosynthesized chromium (CrNPs). These 5-Fu loaded chromium nanoparticles (5-FuCrNPs) are supposed to be effective for antitumor treatments. Several structural methods such as X-ray powder diffraction, scanning and transmission electron microscopy are used to find the structure and size of the nanoparticles.

The determination of the drug loading efficiency was done by direct determination of the unbounded 5-Fu in the filtrate spectrophotometrically at λmax 265 nm.

In Vitro Drug Release of 5-Fu from the formulated CrNPs was performed with a paddle type dissolution test apparatus, SR II, 6 flasks (Hanson Research Co., USA).

A Box-Behnken experimental design (BBD) was utilized to investigate and optimize the formulation parameters of 5-FuCrNPs preparation for maximum LE % and fast drug delivery.

The effects of 5-Fu, CrNPs, and 5-FuCrNPs on the viability of CACO-2 cell line were investigated using MTT cell proliferation assay.

The article needs a major revision for improvement of the presentation. In particular:

  1. The introduction is fragmented and does not smoothly presents the ideas that the authors wish to describe. Please make the introduction more unified and integrated.
  2. Fig.4 has labels for the nanoparticle sizes that are not visible.
  3. Fig.6 caption: “Transform infrared spectra Fourier” to be corrected.
  4. Row 283: “ The effect of changing the 5-Fu conc. on LE% was investigated, at each 5-Fu conc., CrNPs Weight and temperature, using the ANOVA test. LE % values depended on 5-Fu conc., CrNPs Weight and Temperature and were found to range from 55.87±2.03 to 92.11±3.23”, to be rewritten correctly.
  5. What is the reason to use eq. 5, 6 and 7 ? There is no reference who proposed these equations.
  6. In general, sections 3.4 and 3.5 are difficult to read and contain a lot of typing mistakes.
  7. Please append with recent references on the topic of the article related to multi drug effects. Directions in this discussion are given in the following articles: (1) Rational Design of Cancer Nanomedicine: Nanoproperty Integration and Synchronization, Adv. Mater. 2017, 29, 1606628, DOI: 10.1002/adma.201606628 (2) Dual and multi-drug delivery nanoparticles towards neuronal survival and synaptic repair, Neural Regeneration Research, 2017, 12, 886-889, DOI: 10.4103/1673-5374.208546., (3) Light-Activatable Synergistic Therapy of Drug-Resistant Bacteria-Infected Cutaneous Chronic Wounds and Nonhealing Keratitis by Cupriferous Hollow Nanoshells, ACS Nano 2020, 14, 3, 3299–3315, doi.org/10.1021/acsnano.9b08930.

Author Response

Dear Editor of Pharmaceutics journal

First, thank you for your kind concern and cooperation for the opportunity to publish our manuscript in your respectable and high impacted Journal.

We are very excited to have been given the opportunity to revise our manuscript. We carefully considered your comments as well as those offered by the three reviewers. We want to extend our appreciation for taking the time and effort necessary to provide such insightful guidance. The revision, based on the review team’s collective input, includes a number of positive changes. Based on your guidance, we have accordingly modified the manuscript and detailed corrections, changes and/or rebuttals against each point raised are listed below.

We hope that these revisions improve the paper such that you and the reviewers now deem it worthy of publication in Pharmaceuticals. Herein, we explain how we revised the paper based on those comments and recommendations and we offer detailed responses to your comments as well as those of the reviewers. Next, we offer detailed responses of the reviewers’ comments.

Thank you again

Please, find out below the answers to all comments.

Reviewer #1:

  1. The introduction is fragmented and does not smoothly present the ideas that the authors wish to describe. Please make the introduction more unified and integrated.

Response: The introduction has been rewritten, corrected, unified, and integrated

2- Fig.4 has labels for the nanoparticle sizes that are not visible.

Response: We enhanced the resolution of figure 4 so the labels for the nanoparticle sizes are now visible especially with make zoom for the document.

3- Fig.6 caption: “Transform infrared spectra Fourier” to be corrected.

Response: We corrected "Transform infrared spectra Fourier” to "Fourier Transform infrared spectra”

  1. Row 283: “ The effect of changing the 5-Fu conc. on LE% was investigated, at each 5-Fu conc., CrNPs Weight and temperature, using the ANOVA test. LE % values depended on 5-Fu conc., CrNPs Weight and Temperature and were found to range from 55.87±2.03 to 92.11±3.23”, to be rewritten correctly.

Response: It has been rewritten correctly

  1. What is the reason to use eq. 5, 6 and 7 ? There is no reference who proposed these equations.

Response: These equations are regression analysis refers to a method of mathematically sorting out which variables may have an impact. The importance of regression analysis is that it helps determine which factors matter most, which it can ignore, and how those factors interact with each other. The importance of regression analysis lies in the fact that it provides a powerful statistical method that allows our work to examines the relationship between two or more variables of interest.

The benefits of regression analysis are manifold: The regression method of forecasting is used for, as the name implies, forecasting and finding the causal relationship between variables. An important related, almost identical, concept involves the advantages of linear regression, which is the procedure for modeling the value of one variable on the value(s) of one or more other variables…..References were added in line 286(36,42).

  1. In general, sections 3.4 and 3.5 are difficult to read and contain a lot of typing mistakes.

Response: Both sections were revised, rewritten, and corrected. We made it easier to be read. Some statistical analysis data are rigid in their expressions but, specialists are familiar with that.

  1. Please append with recent references on the topic of the article related to multidrug effects. Directions in this discussion are given in the following articles: (1) Rational Design of Cancer Nanomedicine: Nanoproperty Integration and Synchronization, Adv. Mater. 2017, 29, 1606628, DOI: 10.1002/adma.201606628 (2) Dual and multi-drug delivery nanoparticles towards neuronal survival and synaptic repair, Neural Regeneration Research, 2017, 12, 886-889, DOI: 10.4103/1673-5374.208546., (3) Light-Activatable Synergistic Therapy of Drug-Resistant Bacteria-Infected Cutaneous Chronic Wounds and Nonhealing Keratitis by Cupriferous Hollow Nanoshells, ACS Nano 2020, 14, 3, 3299–3315, doi.org/10.1021/acsnano.9b08930.

Response: Thanks for valuable suggestion and articles were added to our manuscript, references: 13, 14 and 16.

Reviewer 2 Report

Many errors and unclear parts in the manuscript that need to be corrected and improved, so my decision is ‘’major revise’’.

  1. The Angstrom unit symbol in lines 121 and 130 is wrong, please correct it in to the correct form.
  2. In line 121, “K.V” and “m.A” need to be changed to “kV” and “mA” to be the correct form.
  3. The numbers in section 2.2.5 should be written as Arabic numerals in the same way as other sections in materials and methods.
  4. Please unify the format, should it be a space between the ‘’%'' and the text or not? For example, ‘’LE %'' and ‘’LE%’’ on lines 283 and 284.
  5. The unit symbol of ‘’ug/ml’’ should be ‘’μg/ml’’.
  6. What do the green and blue points in Figure 2b mean? The experimental points do not seem to have linearity.
  7. In the descriptions of Figures 3 and 4, please rewrite ‘’Scanning micrographs'' and ‘’Transmission electron micrographs ‘’ into ‘’scanning electron microscope’’ and ‘’transmission electron microscope analysis’’. Also, when it comes to the description of common diagrams, we do not use the word ‘’micrographs’’.
  8. Please add 5-Fu analysis to the FTIR analysis chart to confirm that the extra peak in 5-FuCrNPs is contribute by 5-Fu.
  9. It is mentioned in line 319 that "All these results confirm that in order to get higher values of LE%, temperature should be at lower value", why does it result like that? Please make a discussion in the article.
  10. MTT assay is mentioned in Methods and Section 3.6, but the corresponding figure is not seen, please add the experimental result figure of MTT assay.
  11. The concentration of 5-Fu mentioned in line 415 to 417, ‘’Supporting our data, flow cytometry analysis showed that when cells were treated with 5-FuCr-NPs (30 ug/ml), more late apoptotic cells were observed (42.7%) compared to cells treated with the same concentration of 5-Fu (20%).’’ Please clearly indicate how the actual concentration of 5-Fu is used.
  12. In Figure 10, the flow cytometry analysis of CrNPs needs to be added to confirm the cytotoxic effect of CrNPs.
  13. How is the IC50 value of 5-FuCr-NPs and 5-Fu calculated? Please tell in the article, if the IC50 values are calculate by the data that is not given in the current manuscript, please add it in.

Author Response

Dear Editor of Pharmaceutics journal

First, thank you for your kind concern and cooperation for the opportunity to publish our manuscript in your respectable and high impacted Journal.

We are very excited to have been given the opportunity to revise our manuscript. We carefully considered your comments as well as those offered by the three reviewers. We want to extend our appreciation for taking the time and effort necessary to provide such insightful guidance. The revision, based on the review team’s collective input, includes a number of positive changes. Based on your guidance, we have accordingly modified the manuscript and detailed corrections, changes and/or rebuttals against each point raised are listed below.

We hope that these revisions improve the paper such that you and the reviewers now deem it worthy of publication in Pharmaceuticals. Herein, we explain how we revised the paper based on those comments and recommendations and we offer detailed responses to your comments as well as those of the reviewers. Next, we offer detailed responses of the reviewers’ comments.

Thank you again

Please, find out below the answers to all comments.

Reviewer #2:

1- The Angstrom unit symbol in lines 121 and 130 is wrong, please correct it into the correct form.

Response: We wrote the correct symbol of Angstrom unit as Å 

2- In line 121, “K.V” and “m.A” need to be changed to “kV” and “mA” to be the correct form.

Response: We corrected K.V” and “m.A” to “kV” and “mA”

  1. The numbers in section 2.2.5 should be written as Arabic numerals in the same way as other sections in materials and methods.

Response: We corrected that

  1. Please unify the format, should it be a space between the ‘’%'' and the text or not? For example, ‘’LE %'' and ‘’LE%’’ on lines 283 and 284.

Response: We corrected that throughout the manuscript

  1. The unit symbol of ‘’ug/ml’’ should be ‘’μg/ml’’

Response: corrected

6- What do the green and blue points in Figure 2b mean? The experimental points do not seem to have linearity.

Response: This is just a format in the origin program so all points it should be in one color. Thus we uniformed the color of points in the graph. About the points, This is a theoretical approach called Williamson-hall plot obtained from equation 3,4 based on the three values of 2-theta of the investigated CrNPs pattern. We corrected the word experimental in figure 2b to W-H plot

7- In the descriptions of Figures 3 and 4, please rewrite ‘’Scanning micrographs'' and ‘’Transmission electron micrographs ‘’ into ‘’scanning electron microscope’’ and ‘’transmission electron microscope analysis’’. Also, when it comes to the description of common diagrams, we do not use the word ‘’micrographs’’.

Response: We made the required changes.

8-Please add 5-Fu analysis to the FTIR analysis chart to confirm that the extra peak in 5-FuCrNPs is contribute by 5-Fu.

Response: We performed IR spectrum of 5-Fu and added it to Figure 6 for a comparison to confirm that the extra peak in 5-FuCrNPs is contributed with 5-Fu.

  1. It is mentioned in line 319 that "All these results confirm that in order to get higher values of LE%, the temperature should be at lower value", why does it result like that? Please make a discussion in the article.

Response: we discuss that and add a reference (60) Line 312- 320. Our target is to maximize LE%, therefore results and data analysis of BBD optimization suggest the lower value of temperature reflux during preparation to increase adsorption of the drug on NPs surface.

  1. MTT assay is mentioned in Methods and Section 3.6, but the corresponding figure is not seen, please add the experimental result figure of MTT assay.

Reply: We apologize for the missing data. The MTT curve showing the viability of the cells treated with the 5-FuCrNPs and 5-Fu is added to the revised version of the manuscript (Figure 10a).

The paragraph becomes “The cytotoxic effect of 5-FuCrNPs on CACO cells were investigated using MTT assay and compared with that of 5-Fu alone. Both 5-Fu and 5-FuCrNPs exerted a dose-dependent cytotoxic effect (Figure 10a). Interestingly, 5-FuCrNPs markedly decreased the IC50 of 5-Fu/ml from 106 ug/ml to 19.7 ug/ml. Supporting our data, flow cytometry analysis showed that when cells were treated with 5-FuCrNPs (30 ug/ml), more late apoptotic cells were observed (42.7%) compared to cells treated with the same concentration of 5-Fu (20%) or CrNPs alone (11.7%) (Figure 10b-d).”

Figure 10. Viability of the CACO cells treated with different concentrations of 5-FU and 5-FuCrNPs determined by the MTT assay (a). Flow cytometry analysis showing the frequency of apoptotic cells treated with 5-Fu (b), CrNPs (c), and 5-FuCrNPs (d).

  1. The concentration of 5-Fu mentioned in line 415 to 417, ‘’Supporting our data, flow cytometry analysis showed that when cells were treated with 5-FuCrNPs (30 ug/ml), more late apoptotic cells were observed (42.7%) compared to cells treated with the same concentration of 5-Fu (20%).’’ Please clearly indicate how the actual concentration of 5-Fu is used.

Response: Actual concentration was determined from NPs drug content after filtration and determination of unbounded 5-Fu and we mention that in the methodology section

  1. In Figure 10, the flow cytometry analysis of CrNPs needs to be added to confirm the cytotoxic effect of CrNPs.

Reply: In response to the reviewer's comment, we updated figure 10 and included the cytotoxicity of CrNPs (Figure 10c).

  1. How is the IC50 value of 5-FuCr-NPs and 5-Fu calculated? Please tell in the article, if the IC50 values are calculated by the data that is not given in the current manuscript, please add it in.

Reply: We apologize for the missing data. We added the following to the material and methods section. “The IC50, which is the concentration of the sample that caused 50% inhibition of cell viability was determined from the nonlinear regression curve obtained by plotting log concentration of the inhibitor vs. cell viability in percentage (Figure 10a). Each concentration of the inhibitor was tested in triplets. Data were presented as the average IC50 for the tested inhibitory material. Growth curves and regression analysis were performed using the software GraphPad Prism 8.4 (GraphPad Software, San Diego, USA).  “

Reviewer 3 Report

This manuscript reports on the use of chromium nanoparticles prepared for aqueous extracts of Harpullia Pendula 3 and loaded with 5-Fluorouracil in the treatment of colorectal cancer. Overall, the work is exhaustive enough and well conducted. One important point, however, remains elusive throughout the manuscript, i.e. what is the chemical nature of the nanoparticle inorganic core? Are these nanoparticles made by Cr, Cr oxides or what else? I understand that the answer to this question is not easy, but in my opinion the Authors should make the effort in formulating a hypothesis. For instance, he XRD analysis could give some hints with respect to the chemical compound crystallized, but here XRD is just used for linewidth analysis. Moreover, in paragraph 3.3, where the FTIR analysis is presented, the Authors write that “These bio molecules can act as agents of reduction and stabilization”. It seems that here the term reduction is used very vaguely, therefore it should be justified or removed.

Additional remarks and questions are the following:

- The introduction is clear and complete, since it gives convincing accounts of the work design, i.e. its purposes, the choice of the anticancer principle and the choice of the metal for the nanoparticle synthesis. However, in the experimental part no justification is given for the ratio between the metal precursor and the extract amount used in this work. Have other proportions been tried?

- The second part of the abstract is written in a confusing form. The different paragraphs should follow a more schematic line, for instance: “The prepared NPs were characterized for morphology using scanning and transmission 35 electron microscopes (SEM and TEM)”. Should be followed by “The results revealed the formation of a uniform, mono-dispersive, and highly stable CrNPs with mean size of 23 nm. Besides, encapsulation of 5-Fu over CrNPs nanoparticles with higher drug loading efficiency was succeeded with mean size of 29 nm.” Then additional characterization should follow, i.e. “Fourier Transform Infrared (FTIR) and X-ray diffraction pattern (XRD) were also investigated”. Further optimization of the planning should be described: “Box-Behnken Design (BBD) and Response Surface 36 Methodology (RSM) were used to characterize and optimize the formulation factors (5-Fu 37 concentration, CrNPs weight and temperature)” with its corresponding results etc etc. The final part should describe the in vitro experiment and ensuing short conclusions.

- The drug loading efficiency percentage was determined spectrophotometrically. Did the Authors notice a scattering contribution from the chromium nanoparticles to the UV spectrum? And, in this case, how was it subtracted?

- Figure 2b should be eliminated and replaced by a simple averaging of the data. Indeed, using only 3 points can hardly justify a “fitting” procedure

- How were histograms of the diameter distribution shown in figs 4 and 5 built?

- The percentage reported in paragraph 3.4 and their corresponding standard error (e.g. 92.11±3.23) have an unrealistic number of digits. Please, round these value to their physically significant number.

Author Response

Dear Editor of Pharmaceutics journal

First, thank you for your kind concern and cooperation for the opportunity to publish our manuscript in your respectable and high impacted Journal.

We are very excited to have been given the opportunity to revise our manuscript. We carefully considered your comments as well as those offered by the three reviewers. We want to extend our appreciation for taking the time and effort necessary to provide such insightful guidance. The revision, based on the review team’s collective input, includes a number of positive changes. Based on your guidance, we have accordingly modified the manuscript and detailed corrections, changes and/or rebuttals against each point raised are listed below.

We hope that these revisions improve the paper such that you and the reviewers now deem it worthy of publication in Pharmaceuticals. Herein, we explain how we revised the paper based on those comments and recommendations and we offer detailed responses to your comments as well as those of the reviewers. Next, we offer detailed responses of the reviewers’ comments.

Thank you again

Please, find out below the answers to all comments.

Reviewer #3:

  • This manuscript reports on the use of chromium nanoparticles prepared for aqueous extracts of Harpullia Pendula 3 and loaded with 5-Fluorouracil in the treatment of colorectal cancer. Overall, the work is exhaustive enough and well conducted.

  • One important point, however, remains elusive throughout the manuscript, i.e. what is the chemical nature of the nanoparticle inorganic core? Are these nanoparticles made by Cr, Cr oxides or what else? I understand that the answer to this question is not easy, but in my opinion the Authors should make the effort in formulating a hypothesis. For instance, he XRD analysis could give some hints with respect to the chemical compound crystallized, but here XRD is just used for line width analysis. Moreover, in paragraph 3.3, where the FTIR analysis is presented, the Authors write that “These bio molecules can act as agents of reduction and stabilization”. It seems that here the term reduction is used very vaguely, therefore it should be justified or removed.

Response: 1- Thank you very much for your valuable comment. Cr nanoparticles were highly sensitive to air. They were easily oxidized to form Cr2O3 when exposed to air. It is well known that pure chromium metal is easily oxidized to Cr2O3, which is known to be the most thermodynamically stable form of chromium oxide.

To prove that our investigated nanoparticles are CrNPs and not chromium oxide Cr2O3, we deeply check the literature. We found that the obtained XRD pattern in our study is totally different from found in the literature for the stable form of chromium oxide NPs (Cr2O3) as shown in

  • Ceramics International 43 (2017) 2756–2764
  • Microsc Res Tech. 2020;1–14.
  • CODEN: OJCHEG, 2014, Vol. 30, No. (2): Pg. 559-566

Moreover, the Harpullia Pendula Extract which used in the synthesis of CrNPs in our study acts a good reducing and stabilizing agent for CrNPs and prevents it from oxidation to Cr2O3. Furthermore, the obtained pattern of XRD in our study is nearly the same obtained by:

  • Thangavelu Satgurunathan1 & Periyakali Saravana Bhavan1 & Robin David Sherin Joy, Green Synthesis of Chromium Nanoparticles and Their Effects on the Growth of the Prawn Macrobrachium rosenbergii Post-larvae, Biological Trace Element Research (2019) 187:543–552

Thus, we added this reference as ref. 24 in the manuscript to justify our suggestion.

2- About the sentence "These biomolecules can act as agents of reduction and stabilization"

The used extract acts as a good reducing agent to reduce Cr(III) to Cr(0) and also make stabilization for this oxidation state. This behavior is similar to obtained in literature ref. 24 

  • Thangavelu Satgurunathan1 & Periyakali Saravana Bhavan1 & Robin David Sherin Joy, Green Synthesis of Chromium Nanoparticles and Their Effects on the Growth of the Prawn Macrobrachium rosenbergii Post-larvae, Biological Trace Element Research (2019) 187:543–552
  • The introduction is clear and complete since it gives convincing accounts of the work design, i.e. its purposes, the choice of the anticancer principle, and the choice of the metal for the nanoparticle synthesis. However, in the experimental part, no justification is given for the ratio between the metal precursor and the extract amount used in this work. Have other proportions been tried?

Response: Many thanks for your valuable comment which allow us to clarify the efforts we made to select metal we worked on our preliminary studies to adjust H. pendula extract and its aqueous form. We used different Extract: Chromium ratios during NPs preparation. Our findings revealed that there is a non-significant effect of changing this ratio, so we use the 1:1 V/V ratio.

- The second part of the abstract is written in a confusing form. The different paragraphs should follow a more schematic line, for instance: “The prepared NPs were characterized for morphology using scanning and transmission 35 electron microscopes (SEM and TEM)”. Should be followed by “The results revealed the formation of a uniform, mono-dispersive, and highly stable CrNPs with a mean size of 23 nm. Besides, encapsulation of 5-Fu over CrNPs nanoparticles with higher drug loading efficiency was succeeded with a mean size of 29 nm.” Then additional characterization should follow, i.e. “Fourier Transform Infrared (FTIR) and X-ray diffraction pattern (XRD) were also investigated”. Further optimization of the planning should be described: “Box-Behnken Design (BBD) and Response Surface 36 Methodology (RSM) were used to characterize and optimize the formulation factors (5-Fu 37 concentration, CrNPs weight, and temperature)” with its corresponding results etc etc. The final part should describe the in vitro experiment and ensuing short conclusions.

Response: We rearranged the sentences of the abstract according to your instruction

- Figure 2b should be eliminated and replaced by a simple averaging of the data. Indeed, using only 3 points can hardly justify a “fitting” procedure

Response: Figure 2b represents Williamson-hall plot the approximate formulae for size broadening, βL, and strain broadening, βe , vary quite differently with respect to Bragg angle, θ:

βL =KλL cosθ βe =Cε tanθ

One contribution varies as 1/cosθ and the other as tanθ. If both contributions are present then their combined effect should be determined by convolution. The simplification of Williamson and Hall is to assume the convolution is either a simple sum or sum of squares (see the previous discussion on Sources of Peak Broadening within this section). Using the former of these then we get:

βtot = βe + βL = Cε tanθ +KλL cosθ

If we multiply this equation by cosθ we get:

βtot cosθ = Cε sinθ +KλL

and compare this to the standard equation for a straight line (m = slope; c = intercept)

y = mx + c

we see that by plotting βtotcosθ versus sinθ we obtain the strain component from the slope (Cε) and the size component from the intercept (Kλ/L). Such a plot is known as a Williamson-Hall plot and is illustrated schematically below (note that this plot could alternatively be expressed in reciprocal space parameters, β* versus d*):

Our powder x-ray pattern contains only 3 values of θ and thus W-H plot represents the fitting of these three values. moreover, there are different papers that make W-H plot for three points such as:

  • Yasir Rafique, Liqing Pan,  M. Zubair Iqbal, Rafi-ud-din,  Hongmei Qiu,  M. Hassan Farooq, Zhengang Guo, Mujtaba Ellahi. J Nanopart Res (2013) 15:1768.
  • Masoumeh Ghasemi Hajiabadi, Maryam Zamanian, Dariush Souri, Ceramics International 45 (2019) 14084–14089
  • Mohammed Tareque Chowdhury, Md. Abdullah Zubair, Hiroaki Takeda, Kazi Md. Amjad Hussain, and Md. Fakhrul Islam, AIMS Materials Science, 4(5): 1095-1121.

  • How were histograms of the diameter distribution shown in figs 4 and 5 built?

Response: Particle size distribution of the prepared NPs was evaluated using image J Launcher, broken-symmetry soft-ware, version (1.4.3.6.7). This is conducted with using different TEM images with different magnification for the prepared NPs and using the mentioned program, we measured size of each particle. Then calculate the number of particles with the same size and drawing the histogram for these sizes and from fit linear in origin program, we calculated the mean size for the prepared NPs

  • The percentage reported in paragraph 3.4 and their corresponding standard error (e.g. 92.11±3.23) have an unrealistic number of digits. Please, round these value to their physically significant number

Response: we aimed to make it easier and precise for readers to refer these percent to their corresponding formula so that we report them in that format.

Round 2

Reviewer 1 Report

The authors did a substantially improved major revision. I do not have more suggestions. 

Author Response

  1. English language and style are fine/minor spell check required

Response: the manuscript were edited as required and corrections were highlighted in green color

Reviewer 2 Report

All questions have been answered, but a one questions need to be improved

Figure 6. Fourier Transform infrared spectra of the prepared CrNPs, 5-FuCrNPs need baseline correction to make the picture clearer.

Author Response

  • Figure 6. Fourier Transform infrared spectra of the prepared CrNPs, 5-FuCrNPs need baseline correction to make the picture clearer.

Response: We made baseline correction and put all the spectra in one figure with the same scale for comparison